# Mesenchymal Stem Cells: Therapeutic Mechanisms for Stroke

**DOI:** 10.3390/ijms23052550

**Published:** 2022-02-25

**Authors:** Yuchen Zhang, Naijun Dong, Huanle Hong, Jingxuan Qi, Shibo Zhang, Jiao Wang

**Affiliations:** 1School of Life Sciences, Shanghai University, Shanghai 200444, China; yuchen120707@163.com (Y.Z.); dongnaijun@shu.edu.cn (N.D.); honghuanle@shu.edu.cn (H.H.); qijingxuan@outlook.com (J.Q.); zhangshibo1995@shu.edu.cn (S.Z.); 2School of Medicine, Shanghai University, Shanghai 200444, China

**Keywords:** mesenchymal stem cell, stroke, stem cell transplantation, extracellular vehicle

## Abstract

Due to aging of the world’s population, stroke has become increasingly prevalent, leading to a rise in socioeconomic burden. In the recent past, stroke research and treatment have become key scientific issues that need urgent solutions, with a sharp focus on stem cell transplantation, which is known to treat neurodegenerative diseases related to traumatic brain injuries, such as stroke. Indeed, stem cell therapy has brought hope to many stroke patients, both in animal and clinical trials. Mesenchymal stem cells (MSCs) are most commonly utilized in biological medical research, due to their pluripotency and universality. MSCs are often obtained from adipose tissue and bone marrow, and transplanted via intravenous injection. Therefore, this review will discuss the therapeutic mechanisms of MSCs and extracellular vehicles (EVs) secreted by MSCs for stroke, such as in attenuating inflammation through immunomodulation, releasing trophic factors to promote therapeutic effects, inducing angiogenesis, promoting neurogenesis, reducing the infarct volume, and replacing damaged cells.

## 1. Introduction

Stroke is a sudden disorder of cerebral blood circulation that has a rapid onset, with high morbidity and mortality [1]. Due to aging of the world’s population, stroke prevalence and socioeconomic burden are expected to increase [2]. Therefore, the prevention of stroke and treatment of sequelae have become key scientific issues that need urgent attention. Typically, stroke can be categorized into ischemic stroke and hemorrhagic stroke. Ischemic stroke is caused by a blocked blood vessel that reduces blood flow to specific areas of the brain. Hemorrhagic stroke is caused by a blood vessel rupture in the brain, causing bleeding in the brain or subarachnoid space [2,3]. Both ischemic and hemorrhagic stroke are covered in this review.

To treat stroke, the transplantation of mesenchymal stem cells (MSCs) can be used because of their intrinsic ability to play a therapeutic role. MSCs are pluripotent stem cells, with high self-renewal ability and multidirectional differentiation potential, that are capable of differentiating into adipocytes, chondrocytes, osteoblasts, neurons, and glial cells under certain conditions [4,5,6,7]. Moreover, MSCs can be extracted from different parts of the human body, and the isolation and amplification of MSCs is inexpensive. The most commonly studied MSCs are adipose-derived mesenchymal stem cells (ADSCs) and bone marrow-derived mesenchymal stem cells (BMSCs) [8]. ADSCs have several advantages compared to BMSCs. For instance, the proliferative capacity of BMSCs is much lower than that of ADSCs, and adipose tissue (AT) is more abundant than bone marrow (BM). Compared with BMSCs, the yield of ADSCs is not affected by age [9]. In addition, MSCs can be derived from multiple tissues, such as umbilical cord blood (UCB), umbilical cord (UC) [8], tonsils [10], olfactory mucosa [11], and dental tissue [12]. Depending on the cell source, different dosing times may result in different therapeutic effects [13].

There are several delivery methods of MSCs, including intracerebral injection (IC), intracisternal/cerebroventricular injection (ICV), intranasal delivery (IN) [14], and intravascular routes of delivery, such as intravenous injection (IV) or intra-arterial infusion (IA). IV and IC are the most widely used delivery methods in animal and clinical research. Although IV delivery is a simple operation, and is better for large infarctions than IC, few cells can actually reach the target area. On the other hand, IC can deliver drugs directly to the site of injury, but an injection of a certain amount of fluid could lead to increased intracranial hypertension, and the needle may directly cause nerve damage and functional lesions [13] (Figure 1).

## 2. The Mechanisms of MSCs in the Treatment of Stroke

It is important to note that therapies given at the onset of stroke generally aim to reduce the injury, while therapies started days to weeks after stroke tend to promote repair. The following sections will illustrate the mechanisms of MSCs and the main proteins involved in stroke treatment, based on recent studies. This includes attenuating inflammation through immunomodulation, releasing trophic factors to promote therapeutic effects, inducing angiogenesis, promoting neurogenesis, reducing the infarct volume, replacing damaged cells, and secreting extracellular vehicles (EVs), which all play therapeutic roles (Table 1) [4,13,15,16].

### 2.1. Attenuate Inflammation through Immunomodulation

After stroke, microglia cells switch from a resting form to an activated state and adopt a phagocytic phenotype to secrete pro-inflammatory cytokines [34]. When MSCs were transplanted into animal models, they produced inflammatory mediators and influenced the expression of cytokines [16]. On the one hand, MSCs could increase the secretion of anti-inflammatory cytokines, such as interleukin-4 (IL-4), interleukin-10 (IL-10), and tumor necrosis factor β (TNF-β). On the other hand, MSCs could reduce the expression of pro-inflammatory cytokines, including interleukin-1 (IL-1), interferon γ (IFN-γ), tumor necrosis factor α (TNF-α), and membrane cofactor protein-1 (MCP-1) [17,18]. By regulating these cytokines, MSCs affected several pathways involved in immune cells and immune responses, to reduce inflammation [35].

Studies showed that TNF-α and IFN-γ were the main pro-inflammatory cytokines. Prostaglandin E2 (PGE2) was their related mediator, which was proven to be an important mediator of MSCs, for inhibiting the immune response and inflammation by regulating immunity, inhibiting T-cell proliferation, and changing T-cell differentiation [19]. The level of PGE2 was demonstrated to decrease after stroke and increase after MSC transplantation. Subsequently, both the secretion of TNF-α in dendritic cells and the secretion of IFN-γ in T helper 1 and natural killer (NK) cells decreased [36,37]. As a result, the density of TNF-α decreased significantly, indicating that MSCs reduced stroke-induced neuroinflammation.

In addition to the main pro-inflammatory factors mentioned above, MSCs also regulated the expression of late pro-inflammatory cytokines, which played a therapeutic role. High-mobility group box 1 (HMGB1) was a late pro-inflammatory cytokine, and the release of HMGB1 could be induced by early pro-inflammatory cytokines, such as TNF-α [20]. HMGB1 exacerbated inflammation caused by early pro-inflammatory cytokines. The injection of BMSCs significantly reduced the expression of HMGB1, and the receptor for advanced glycation end-products (RAGE), which was a receptor of HMGB1, could interact with HMGB1 and form a positive feedback loop for HMGB1-mediated inflammation. Ultimately, BMSCs reduced the inflammatory response caused by HMGB [21].

Based on a large number of animal studies investigating stroke, there is growing interest in the anti-inflammatory potential of MSCs for the treatment of neuroinflammatory diseases. It has been found that MSCs can regulate the immune response to play an anti-inflammatory function, by secreting a variety of immune factors, and these regulatory mechanisms can be used as potential therapeutic approaches. However, MSCs may induce certain pro-inflammatory responses through several signaling pathways, which calls for more studies. These mechanisms for inhibiting inflammation will be crucial for future research to improve the efficacy of MSCs-based therapies for neurological diseases.

### 2.2. Release Trophic Factors to Promote Therapeutic Effects

MSCs released, or stimulated the release of, trophic factors that were related to the mechanisms of stroke treatment. These trophic factors not only included neurotrophic factors, such as brain-derived neurotrophic factor (BDNF) and glial cell line-derived neurotrophic factor (GDNF), but also growth factors, such as nerve growth factor (NGF), vascular endothelial growth factor (VEGF), and platelet-derived growth factor (PDGF).

These trophic factors secreted by MSCs could reduce the infarct volume, prevent neuron apoptosis, increase neuron proliferation, and induce angiogenesis. After transplantation, MSCs migrated from the vascular system outside the lesion to the area of infarction, to reduce the infarct volume by secreting BDNF [22]. Both MSCs and BDNF gene-modified MSCs could reduce the infarct volume and improve neurogenesis, but the effect of the latter was more obvious because transplanting MSCs carrying the BDNF gene maintained high levels of BDNF during the critical period after stroke [38]. Additionally, transplanted MSCs carrying the GDNF gene also reduced the infarct volume, and the effect was similar to MSCs carrying the BDNF gene [23]. In addition, the overexpression of the BDNF gene could direct the MSC into the neural differentiation pathway [7]. In other aspects, the function of NGF was to prevent neuron apoptosis and to increase neuron proliferation [24], while VEGF enhanced angiogenesis [25], and PDGF promoted cell migration, the growth of primary cortical neurons, angiogenesis and axon growth, as well as the inhibition of neuroinflammation [26,27,28].

It is noteworthy that the amount of these trophic factors released by MSCs was related to the passage MSCs cultured, the source of MSCs, and other influence factors. The application of ex vivo-cultured human MSCs in vitro, earlier passage or later passage, in a rat stroke model, showed that the behavioral recovery and neurogenesis were more pronounced in rats receiving the earlier passage MSCs than those receiving the later passage MSCs. This is mainly because the trophic factors secreted by MSCs, such as GDNF, NGF, and VEGF, were essentially higher in earlier passage MSC-treated brains than in later passage MSC-treated brains [39]. Additionally, the source of MSC also affected its therapeutic properties, particularly considering that the amount of VEGF secreted by ADSCs was significantly higher than that of BMSCs [9].

In summary, MSCs played diverse therapeutic roles by secreting a variety of trophic factors. In the future, gene modification could be the main research direction, in order to study how the therapeutic effects of MSCs can be enhanced. In addition to modifying a single gene, several genes, with similar effects, could also be modified together to study therapeutic effects. However, the adverse effects of excessive trophic factors on neurons should be avoided [40].

### 2.3. Induce Angiogenesis

After the transplantation of MSCs, the expression of angiogenin and the vascular density of ischemic brain tissue were significantly increased. It was found that in damaged blood vessels, MSCs showed high expression of several factors associated with angiogenesis and arterial density, such as angiogenin-1 (Ang1), tyrosine-protein kinase receptor (Tie2), VEGF, and VEGF receptor 2 (Flk1). Tie2 was the receptor of Ang1 and Flk1 was the receptor of VEGF. The VEGF/Flk1 system and Ang1/Tie2 system contributed to the enhancement of angiogenesis [29]. The expression of VEGF, Ang1, and Tie2 was significantly increased in astrocytes and endothelial cells co-cultured with BMSCs [29]. In addition, BMSC-conditioned medium treatment by a single intravenous injection after stroke, in type 2 diabetes mellitus (T2DM) rats, also increased Ang1 and Tie2 expression [41].

To make the treatment of MSCs more effective, some researchers used gene-modified MSCs, such as Ang1 gene-modified hMSCs (Ang-hMSCs). The results showed that the grafted Ang-hMSCs had more neovascularization and local cerebral blood flow in marginal regions [42]. Another study showed that the intravenous administration of C-C motif chemokine ligand 2 (CCL2)-overexpressing UCB-MSCs increased angiogenesis, due to cell migration to brain regions with higher CCL2 receptor expression, which promotes subsequent endogenous brain repair [43].

To this end, MSCs have pro-angiogenic effects in the treatment of cerebral apoplexy, mainly through the mechanism of promoting growth factor secretion and binding chemokines. The regeneration of blood vessels in the brain is crucial for stroke patients, so the relevant mechanisms need to be thoroughly studied to prevent aggravation of the patients’ conditions and other adverse consequences in the future.

### 2.4. Promote Neurogenesis

The first way in which MSC transplantation can promote neurogenesis involves enhancing the proliferation of endogenous neural cells. In short, the enhancement of neuroplasticity and neural proliferation from the ischemic boundary zone (IBZ) was among the mechanisms by which BMSC therapy could lead to functional recovery after stroke. The BrdU (+) cells around the infarct area were significantly increased after MSC treatment, which could indicate the proliferation of more cells [44].

Indeed, MSCs promoted the growth of axons, synapses, and myelin of IBZ, thus improving neural function. BMSCs significantly increased the axonal growth of primary cortical neurons [45]. After MSC treatment in stroke mice, there was a significant increase in the expression of axonal growth-associated proteins, and a significant decrease in the expression of axonal growth-inhibiting proteins [30]. In terms of the promotion of synaptic plasticity after stroke [46], MSC-generated exosome (Exos) treatment increased IBZ synaptophysin, thereby increasing synaptic remodeling and synaptic plasticity [47]. In addition, MSCs could induce myelination. With regards to inducing myelination, the number of oligodendrocyte cells and oligodendrocyte progenitor cells in the IBZ of rats undergoing MSC-generated Exos treatment was increased. The oligodendrocyte cells contributed to promoting myelin growth and protecting myelin from damage [48].

Furthermore, intravenously injected MSCs improved the neurological function recovery of the blood–brain barrier (BBB). In this case, MSCs increased the expression levels of collagen IV and the tight junction protein zonula occludens-1 (ZO-1) expression levels in the damaged brain of intracerebral hemorrhage rats, to decrease BBB disruption and neuronal loss [31,32]. Meanwhile, T2DM-MSC-Exo treatment of T2DM stroke rats increased tight junction protein ZO-1 expression and improved BBB integrity [48]. On the same note, in T2DM rats with stroke, BMSC transplantation significantly reduced BBB leakage and promoted neurorestorative effects [21,49], including neuroblast migration and white matter remodeling, specifically by increasing doublecortin, axon, myelin and neurofilament density [26].

The second way in which MSC transplantation can promote neurogenesis is by protecting new growing cells from the pathogenic environment, to prevent their death. Of course, reducing neuroinflammation prevented cell death. A recent study showed that MSC spheroid-loaded collagen hydrogels could not only promote neurogenesis, but also suppress the neuronal inflammatory response by creating micro-environmental niches to reduce neuroinflammation. MSC spheroid-loaded collagen hydrogels played a therapeutic role through three up-regulated signals related to cell communication, followed by the activation of a signaling pathway related to neuroactive ligand–receptor interactions, and, lastly, by up-regulating the PI3K-Akt signaling pathway, which increased the expression of proteins related to neurogenesis and neuroprotection [18].

The promotion of neurogenesis is an important mechanism for MSC therapy, to enable patients to recover from stroke through the repair of damaged sites. However, due to the limited regeneration ability of neural stem cells (NSCs) and the complex physiological environment, their repair effect is not ideal. For that reason, MSCs could be used to promote the differentiation of NSCs into neurons by the production of different kinds of nutritional factors and anti-apoptotic molecules. Future research should, therefore, focus on the development of MSC cell therapies related to NSCs, to promote recovery of the nervous system.

### 2.5. Reduce Infarct Volume

The infarct volume, reduced by MSCs, is related to the source of cells, species studied, and timing of MSC injection after stroke. In vivo experiments showed that ADSCs, BMSCs, as well as UC-MSCs [44] could reduce the infarct volume in rats. ADSC-treated mice had smaller infarcts than BMSC-treated mice [9].

ADSCs and ADSC-derived Exos could limit the size of cerebral infarction and enhance neurological recovery, without tumor development. After ADSC-derived Exos treatment, acute brain swelling was reduced and cerebral atrophy, induced by infarction, was reduced. In addition, no immune response was found in rodents administered xenogenic ADSCs/ADSC-derived Exos for acute ischemic stroke [50]. Because of the therapeutic effect of MSCs, combination therapy of MSCs and recombinant tissue plasminogen activator (rtPA) could improve the outcome of rtPA treatment. In fact, the infarct volume in intracerebral hemorrhage rats treated with both MSCs intravenously and rtPA was significantly reduced, compared with the two treatments individually. Should bleeding complications occur after intravenous rtPA treatment, injected MSCs may inhibit endothelial dysfunction, thus inhibiting bleeding events and promoting functional outcomes [51].

### 2.6. Replace Damaged Cells

The transplantation of MSCs may contribute to brain tissue regeneration, partly through their potential to differentiate into neurons and glial cells under appropriate conditions. Two factors are necessary for MSC differentiation into neurons. The first is that MSCs express nestin, which is a marker of corresponding extrinsic signals in MSCs. The second is a direct cell–cell interaction between MSCs and neurons, which allows the integration of extrinsic signals [52].

For decades, studies have shown that MSCs can differentiate into neurons and glial cells. In earlier studies, MSCs were successfully induced to nestin (+) neurospheres, and cultured in a medium containing epidermal growth factor (EGF) and basic fibroblast growth factor (bFGF). After withdrawal of the mitogens in the medium, these neurospheres could differentiate into neurofilament (+) neurons or glial fibrillary acidic protein (GFAP) (+) glia cells [53]. When ADSCs differentiated into neurons or glial cells, the ADSCs expressed molecular markers for neurons, such as neuron-specific β-tubulin (Tuj-1), neuron-specific enolase (NSE), microtubule association protein-2 (MAP2), and neuronal nuclei (NeuN), or expressed astrocyte markers, such as GFAP, and oligodendrocyte markers, such as 2′,3′-cyclic-nucleotide 3′-phosphodiesterase (CNPase) [9].

More recent research has described a referential neurobasal medium in which MSCs could differentiate into neurons. MSCs were first cultured in the medium with fibroblast growth factor (FGF) and bFGF. After 9 days, the medium was supplemented with BDNF and GDNF. The cells were then cultured in this medium for another 3 days [11]. In light of this, future studies could aim to continue to improve this medium, to obtain a better culture scheme to promote the differentiation of MSCs.

### 2.7. Play a Therapeutic Role through EVs

Based on the paracrine mechanism of MSCs, EVs with genetic regulatory information can be secreted by MSCs, becoming a new strategy for cell-free therapy. Compared with the EVs of neural progenitor cells, the EVs of MSCs had a better therapeutic effect [54]. The EVs of MSCs include Exos and microparticles (MPs). Exos have the advantages of low toxicity and immunogenicity, biodegradation, the ability to encapsulate endogenous bioactive molecules, and the ability to cross the BBB [55]. Studies are presently focusing on the function of the Exos and MPs isolated from BMSCs, ADSCs, and, sometimes, UC-MSCs [16].

Since the active substances in EVs include some proteins and microRNAs (miRNAs), the EVs of MSCs exerted various regulatory mechanisms through these miRNAs. This way, miRNAs could be activated by proteins and bind to their corresponding targets, thus affecting downstream signals and producing therapeutic effects [56]. After the administration of MSC-EVs, B lymphocytes, T lymphocytes, and NK cells were reduced, providing a suitable anti-inflammatory environment for brain remodeling [57]. In addition, MSC-EVs significantly decreased the infarct volume and neuronal injury [58]. They could also increase the migration and tube formation of endothelial cells to promote angiogenesis [59]. Additionally, the EVs from two different BMSC lineages delivered to mice were reported to improve nerve injury and long-term neuroprotection, associated with enhanced neurogenesis [57]. Nowadays, besides the treatment of stroke as mentioned above, MSC-EVs can also be used for coronavirus disease 2019 (COVID-19) therapy. However, the safety and efficacy of MSC-EVs are still under evaluation [60].

## 3. Adverse Effects

Many preclinical studies have comprehensively investigated the adverse effects of MSC therapy. It was reported that MSCs can stimulate tumor growth [61], although the increased risk of tumor formation has never been demonstrated in humans [62]. In actuality, the intravenous administration of autologous BMSCs did not result in any significant adverse events, but there were several slight adverse events. Recent clinical trials showed that MSC therapy was safe, but had no functional improvement [63,64]. A study showed that mild itching at the point of injection was observed in one patient, mild fever and nausea in one patient, and mild loss of appetite in one patient [65]. In another study, autologous MSCs, grafted intravenously, showed no adverse effects and promoted neural recovery [66]. Many studies have used unmatched allogeneic MSCs, and no acute infusion toxicity has been reported, but the administration of necrotic cells or cell by-products may increase immunogenicity in low-viability MSCs [67].

In general, the intravenous injection of in vitro-cultured autologous MSCs is a safe and feasible treatment for ischemic stroke, because it avoids immune reactions [66]; however, its therapeutic effect needs to be improved. The methods to improve the efficacy of stem cell therapy for ischemic stroke have been widely explored in recent years. In many types of stem cells, sub-lethal hypoxic exposure significantly increased cell tolerance and regeneration characteristics. Pretreated stem cells showed better cell survival, increased neuronal differentiation, enhanced paracrine effect, increased nutritional support, and improved homing to the lesion site [68].

## 4. Conclusions

In summary, a variety of therapeutic mechanisms and effects of MSCs have been revealed, as illustrated in Figure 2. In essence, MSC therapy has been proposed as a future biologic cell therapy for tissue repair and regeneration, particularly because of its pluripotency and universality [69]. Despite MSCs having received special attention as a promising candidate for stroke treatment, there are still many unanswered questions, such as the drug delivery route and cell dosage to be used [70]. Although stem cell-based therapies can improve the neurological deficits and activities of daily living in patients with ischemic stroke, the effect is not enough to produce a significant improvement for patients [63]. Besides, the clinical trials of stem cell therapy for ischemic stroke are still in their early stages. It is also important to mention that obtaining a large number of MSCs from the tissue of a stroke patient usually takes a long time, often between two and three weeks [71]. Furthermore, in most of the studies, cases in the stem cell group were very limited [72].

The combination therapy of MSCs with drugs, or with gene modification, is the focus of research in the field. For instance, MSC therapy combined with royal jelly significantly reduced the infarcted volumes, cerebral edema, and serum pro-inflammatory cytokine levels [73]. The administration of hADSC-loaded P5 peptide to post-stroke rats created conditions that supported drug-loaded hADSC survival, and hADSC loaded with p5 peptide reduced the number of inflammatory cells around the lesion, but did not increase the vascular density around the infarction area [74]. Gene-modified MSCs can express biologically active gene products, and, therefore, serve as effective vectors for therapeutic gene transfer to the brain. Thus, gene-modified MSC therapy is a useful approach for the treatment of stroke. For instance, BDNF, GDNF, ciliary neurotrophic factor, and neurotrophin-3 genes can be transfected into MSCs by a fiber-mutant adenovirus vector, increasing trophic factors and cytokines needed for MSCs to attenuate inflammation, induce angiogenesis, reduce the infarct volume, and other functions [23].

In order to minimize the immune response to MSC allotransplantation and reduce the antibodies produced by allogeneic reactions, subsequent studies could focus on the immunogenicity, survival, and specific effects of mesenchymal stem cell therapy. In addition to developing new treatments to reduce side effects and immune rejection, new indicators of therapeutic efficacy could also be explored. In addition, multiple endpoint criteria are also required, and both histology and behavior results should be evaluated.

In conclusion, this paper summarizes the mechanisms of MSCs in the treatment of stroke, such as attenuating inflammation through immunomodulation, releasing trophic factors to promote therapeutic effects, inducing angiogenesis, promoting neurogenesis, reducing the infarct volume, and replacing damaged cells. In addition, this paper describes the proteins and signaling pathways related to the therapeutic effects of MSCs, and many new ideas on the treatment of stroke by MSCs are proposed, which is expected to be enlightening and helpful for future research.

## Figures and Tables

**Figure 1 ijms-23-02550-f001:**
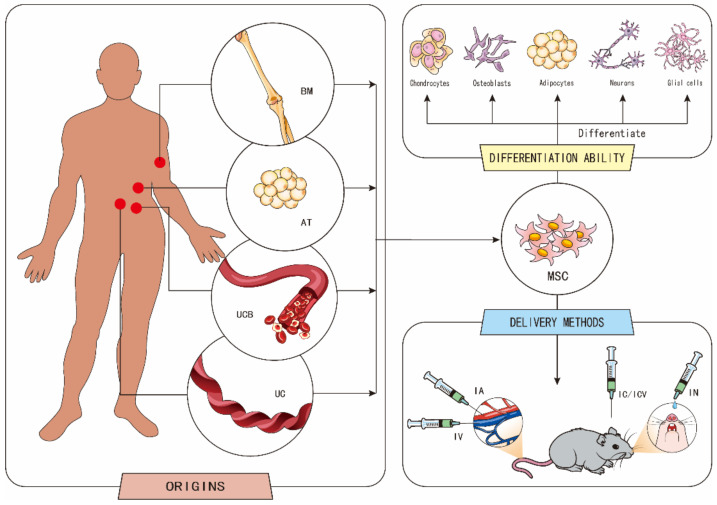
The origins, differentiation ability, and delivery methods of mesenchymal stem cells (MSCs). MSCs are mainly derived from adipose tissue (AT), bone marrow (BM), umbilical cord blood (UCB), and umbilical cord (UC). MSCs have the potential to differentiate into adipocytes, chondrocytes, osteoblasts, neurons, and glial cells. MSCs can be delivered by intracerebral injection (IC), intracisternal/cerebroventricular injection (ICV), intranasal delivery (IN), and intravascular routes of delivery, such as intravenous injection (IV) or intra-arterial infusion (IA).

**Figure 2 ijms-23-02550-f002:**
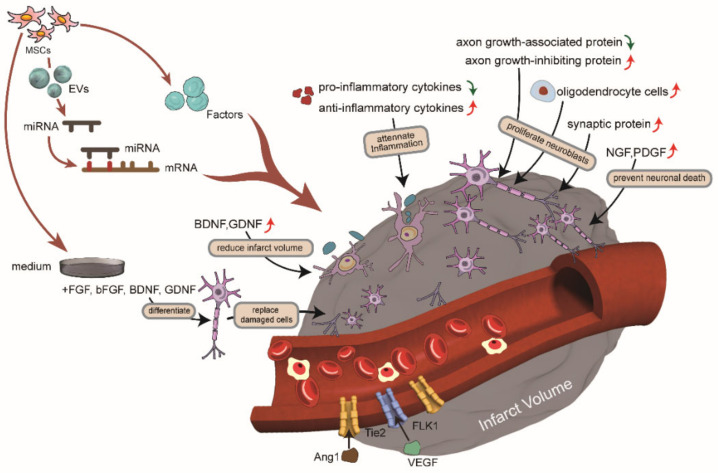
The therapeutic mechanisms of mesenchymal stem cells (MSCs) and extracellular vehicles (EVs) secreted by MSCs for stroke. After stroke, the vessel was infarct, and neurons were damaged. Transplantation of MSCs could play a therapeutic role. MSCs secreted factors and EVs to attenuate inflammation through immunomodulation, release trophic factors to promote therapeutic effects, induce angiogenesis, promote neurogenesis, reduce infarct volume, and replace damaged cells.

**Table 1 ijms-23-02550-t001:** Proteins involved in the therapeutic mechanism of MSCs.

Therapeutic Benefits	Proteins	Mechanisms
Attenuate inflammation through immunomodulation	IL-1, IFN-γ, TNF-α, MCP-1	Decreased pro-inflammatory cytokines to attenuate inflammation [17,18]
	IL-4, IL-10, TNF-β	Increased anti-inflammatory cytokines to attenuate inflammation [17,18]
	PGE2	Mediated the expression of TNF-α and IFN-γ [19]
	HMGB1	Late pro-inflammatory cytokine [20,21]
Release trophic factors to promote therapeutic effects	BDNF	Promoted neurological recovery [22] and directed differentiation of MSCs [7]
	GDNF	Reduced infarct volume [23]
	NGF	Prevented neuron apoptosis and increased neuron proliferation [24]
	VEGF	Induced angiogenesis [25]
	PDGF	Promoted the migration of cells, promoted the growth of primary cortical neurons, inhibited neuroinflammation, and promoted angiogenesis and axon growth [26,27,28]
Induce angiogenesis	Ang1 and tyrosine protein kinase receptor Tie-2	Increased these proteins to increase blood vessel density at the site of vascular injury [29]
	VEGF and VEGF receptor 2 (Flk1)	
Proliferate neuroblasts	Axonal growth-associated proteins and axonal growth-inhibiting proteins	Increased axonal growth-associated proteins and decreased axonal growth-inhibiting proteins to promote axonal growth [30]
	Collagen IV and tight junction protein ZO-1	Increased these proteins to decrease BBB disruption and neuronal loss [31,32]
	p53 protein	Reduced the activity of p53 protein to decrease neuron apoptosis [33]
Replace damaged cells	MAP2 and NeuN	Differentiated into new neurons to replace damaged neurons [9]
	GFAP and CNPase	Differentiated into new glial cells to replace damaged glial cells [9]

## Data Availability

Not applicable.

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
