# Peer review of "Mesenchymal Stem Cells: Therapeutic Mechanisms for Stroke"

_ijms, 2022, doi:10.3390/ijms23052550_

Round 1

Reviewer 1 Report

The Ms: Mesenchymal Stem Cells Therapeutic Mechanisms for Stroke, by Zhang et al is an interesting and relevant Review on MSC administration as therapeutic strategy against stroke.

The authors have experience in the theme, the review is well supported by relevant and contemporary literature. Thus, the MS may contribute to the international development of the theme.

However, I have some comments that in my opinion the authors must address.

MAIN TOPICS

  1. The authors inform in the Introduction Section that strokes may be ischemic or hemorrhagic. However, they do not specify if this review refers to ischemic or both strokes. This topic has to be clarified.
  2. The literature is well cited. However, very frequently, it is not clear if the cited study refers to a human or an experimental study. I recommend addressing this topic
  3. The “Adverse Effects” of MSCs treatment should be included (detailing and discussing the adverse effects) in a specific topic with this title, and not in the final Discussion Section.
  4. Although the title of the MS is on “MSCs therapy”, the EVs/exosomes released from MSCs are cited through the text. For me, it was not clear if the authors refer to administration of EVs/Exosomes previously released from MSCs or to EVs/Exosomes released into the parenchyma after MSCs administration. The authors have to clarify this topic.
  5. Concerning the delivery methods, the authors did not cite the Intranasal Delivery. There are references on this route, including review. The authors should include this route as an option, citing a REF.

MINOR TOPICS

  1. In Table 1., which refers to MSCs therapeutic mechanisms, there are some confused topics: 1) “Pro-inflammatory” cytokines should be cited, emphasizing that the therapeutic mechanism involves its “decreased” levels; 2) some concepts as “Reduced infarct volume” is not a true “mechanism”.
  2. Lines 137-138. This information should include a REF.
  3. Line 277: “the effect is not enough to produce a significant change for patient” clearly include a REF.

Author Response

Dear Editors and Referees,

We are very grateful for this chance to modify our manuscript. We thank you for the constructive criticism and valuable comments. We have carefully read our review and revised it.

MAIN TOPICS

1. The authors inform in the Introduction Section that strokes may be ischemic or hemorrhagic. However, they do not specify if this review refers to ischemic or both strokes. This topic has to be clarified.

Response 1: Both ischemic and hemorrhagic strokes are covered in this article, although most of the reviewed studies are inclined towards the most common ischemic stroke. The studies that are particularly on hemorrhagic stroke are duly noted such as in line 191 and 228.

We have included a clarification about addressing this comment in lines 29-30 of introduction.

2. The literature is well cited. However, very frequently, it is not clear if the cited study refers to a human or an experimental study. I recommend addressing this topic

Response 2: The studies in this review refer to experimental studies except the “Adverse Effects” section. The studies cited were conducted on animal models and did not involve human experiments except the “Adverse Effects” section.

3. The “Adverse Effects” of MSCs treatment should be included (detailing and discussing the adverse effects) in a specific topic with this title, and not in the final Discussion Section.

Response 3: Thank you for your suggestions. The “Adverse Effects” is included in a specific topic according to your suggestion, and the content is supplemented in line 278-298.

3. Adverse effects

Many preclinical studies have comprehensively investigated the adverse effects of MSC therapy. It was reported that MSCs can stimulate tumor growth[61], although the increased risk of tumor formation has never been demonstrated in humans[62]. Actually, intravenous administration of autologous BMSCs did not result in any significant adverse events but there were several slight adverse events. Recent clinical trials showed that MSC therapy was safe but had no functional improvement[63, 64]. A study showed that mild itching at the point of injection was observed in one patient, mild fever and nausea in one patient, and mild loss of appetite in one patient[65]. In another study, autologous MSCs grafted intravenously showed no adverse effects and promoted neural recovery[66]. Many studies have used unmatched allogeneic MSCs and no acute infusion toxicity has been reported, but administration of necrotic cells or cell by-products may increase immunogenicity in low-viability MSCs[67].

In general, intravenous injection of in vitro cultured autologous MSCs is a safe and feasible treatment for ischemic stroke because it avoids immune reactions[66]. But its therapeutic effect needs to be improved. The methods to improve the efficacy of stem cell therapy for ischemic stroke have been widely explored in recent years. In many types of stem cells, sub-lethal hypoxic exposure significantly increased cell tolerance and regeneration characteristics. Pretreated stem cells showed better cell survival, increased neuronal differentiation, enhanced paracrine effect, increased nutritional support, and improved homing to the lesion site[68].

61. Zhang, L. N.;  Zhang, D. D.;  Yang, L.;  Gu, Y. X.;  Zuo, Q. P.;  Wang, H. Y.;  Xu, J.; Liu, D. X., Roles of cell fusion between mesenchymal stromal/stem cells and malignant cells in tumor growth and metastasis. The FEBS journal 2021,288(5), 1447-1456.

62. Jaillard, A.;  Hommel, M.;  Moisan, A.;  Zeffiro, T. A.;  Favre-Wiki, I. M.;  Barbieux-Guillot, M.;  Vadot, W.;  Marcel, S.;  Lamalle, L.;  Grand, S.; Detante, O., Autologous Mesenchymal Stem Cells Improve Motor Recovery in Subacute Ischemic Stroke: a Randomized Clinical Trial. Transl Stroke Res 2020,11(5), 910-923.

63. Borlongan, C. V., Concise Review: Stem Cell Therapy for Stroke Patients: Are We There Yet? Stem cells translational medicine 2019,8(9), 983-988.

64. Steinberg, G. K.;  Kondziolka, D.;  Wechsler, L. R.;  Lunsford, L. D.;  Kim, A. S.;  Johnson, J. N.;  Bates, D.;  Poggio, G.;  Case, C.;  McGrogan, M.;  Yankee, E. W.; Schwartz, N. E., Two-year safety and clinical outcomes in chronic ischemic stroke patients after implantation of modified bone marrow-derived mesenchymal stem cells (SB623): a phase 1/2a study. Journal of neurosurgery 2018, 1-11.

65. Honmou, O.;  Houkin, K.;  Matsunaga, T.;  Niitsu, Y.;  Ishiai, S.;  Onodera, R.;  Waxman, S. G.; Kocsis, J. D., Intravenous administration of auto serum-expanded autologous mesenchymal stem cells in stroke. Brain : a journal of neurology 2011,134(Pt 6), 1790-807.

66. Bang, O. Y.;  Lee, J. S.;  Lee, P. H.; Lee, G., Autologous mesenchymal stem cell transplantation in stroke patients. Annals of neurology 2005,57(6), 874-82.

67. Lalu, M. M.;  McIntyre, L.;  Pugliese, C.;  Fergusson, D.;  Winston, B. W.;  Marshall, J. C.;  Granton, J.; Stewart, D. J., Safety of cell therapy with mesenchymal stromal cells (SafeCell): a systematic review and meta-analysis of clinical trials. PloS one 2012,7(10), e47559.

68. Yu, S. P.;  Wei, Z.; Wei, L., Preconditioning Strategy in Stem Cell Transplantation Therapy. Translational Stroke Research 2013,4(1), 76-88.

4. Although the title of the MS is on “MSCs therapy”, the EVs/exosomes released from MSCs are cited through the text. For me, it was not clear if the authors refer to administration of EVs/Exosomes previously released from MSCs or to EVs/Exosomes released into the parenchyma after MSCs administration. The authors have to clarify this topic.

Response 4: As we wrote in line 257-258 of Paragraph 2.7, EVs can be secreted by MSCs, becoming a new strategy for cell-free therapy[1]. The co-existence of MSCs is not required for EVs to play a therapeutic role. EVs are previously released from MSCs before the administration.

1.  Zheng, X.;  Zhang, L.;  Kuang, Y.;  Venkataramani, V.;  Jin, F.;  Hein, K.;  Zafeiriou, M. P.;  Lenz, C.;  Moebius, W.;  Kilic, E.;  Hermann, D. M.;  Weber, M. S.;  Urlaub, H.;  Zimmermann, W. H.;  Bähr, M.; Doeppner, T. R., Extracellular Vesicles Derived from Neural Progenitor Cells--a Preclinical Evaluation for Stroke Treatment in Mice. Transl Stroke Res 2021,12 (1), 185-203.

5. Concerning the delivery methods, the authors did not cite the Intranasal Delivery. There are references on this route, including review. The authors should include this route as an option, citing a REF.

Response 5: Thank you for your suggestion. We have included this route as an option and the content is supplemented in line 46 and Figure 1.

MINOR TOPICS

1. In Table 1., which refers to MSCs therapeutic mechanisms, there are some confused topics: 1) “Pro-inflammatory” cytokines should be cited, emphasizing that the therapeutic mechanism involves its “decreased” levels; 2) some concepts as “Reduced infarct volume” is not a true “mechanism”.

Response 1: 1) "Pro-inflammatory" cytokines are listed in "Proteins" and several “Mechanisms” in Table 1. have been replenished; 2) the “Reduced infarct volume” in Table 1 has been replaced with “Promoted neurological recovery”.

2. Lines 137-138. This information should include a REF.

Response 2: Thank you very much for your comment, Lines 142-143 is supplemented as you suggested.

3. Line 277: “the effect is not enough to produce a significant change for patient” clearly include a REF.

Response 3: Thank you very much for your comment, Lines 305-307 is supplemented as you suggested.

Besides your comments and suggestions, we have also made further improvements to this review as shown below. Thank you for your useful comments and corrections.

  1. Line 88: ”dendritic cells 1” has been modified to ”dendritic cells”.
  2. Line 160: ”CCR2” has been modified to ”CCL2 receptor”.
  3. Line 176-178: ”After MSCs treatment in stroke mice, there was significant increase and inhibition in the expression of the axonal growth-associated protein and the axonal growth-inhibiting proteins, respectively” has been modified to ”After MSCs treatment in stroke mice, there were a significant increase in the expression of the axonal growth-associated proteins and a significant decrease in the expression of the axonal growth-inhibiting proteins”.
  4. Line 183: ”Third” has been modified to ”Besides”.
  5. Line 196-197: “the density of bicocorticoids, axons, myelin sheaths, and nerve filaments” has been modified to ”doublecortin, axon, myelin and neurofilament density'.
  6. Line 202: ”By” has been modified to ”by”.
  7. Line 203: ”First” has been removed.
  8. Line 213: ”research” has been modified to ”researches”.
  9. Line 262-263: ”the function and differentiation” have been modified to ”the function”.
  10. Line 366: ”N.-J.D., H.-L.H., S.-B.Z.” has been modified to ”N.-J.D., H.-L.H., S.-B.Z., J.W.”.
  11. Line 369: “Basic Research Program of Shanghai (20JC1412200)" has been put in the first place.

We hope that these changes have made the manuscript acceptable for publication. We owe a great deal to the editor for their patience and assistance in helping us to improve the paper.

Best regards,

Jiao Wang

Reviewer 2 Report

Dear Authors,

this manuscript is very interesting, but there are some suggested revisions that should be address; it is possible see my comments below.

Please specify all acronyms reported in figure 1

Please specify aronyms in figure 2

Changes the name of the Figure2.... in Figure 2

Changes mistake in the page 8, line 312

Author Response

Dear Editors and Referees,

We are very grateful for this chance to modify our manuscript. We thank you for the constructive criticism and valuable comments. We have carefully read our review and revised it.

1. Please specify all acronyms reported in figure 1

Response 1: Thank you for your suggestion. All acronyms reported in Figure 1 were specified accordingly in figure legends.

2.Please specify aronyms in figure 2

Response 2: Thanks again for your suggestion. All acronyms reported in Figure 2 were specified in figure legends. Figure 2 is a summary of this review and all the acronyms of the proteins in Figure 2 have been mentioned several times in the article, so it is not explained in figure legends.

3.Changes the name of the Figure2.... in Figure 2

Response 3: We have modified it according to your suggestion in line 301.

4.Changes mistake in the page 8, line 312

Response 4: We have modified it according to your suggestion in line 342.

Besides your comments and suggestions, we have also made further improvements to this review as shown below. Thank you for your useful comments and corrections.

  1. Line 88: ”dendritic cells 1” has been modified to ”dendritic cells”.
  2. Line 160: ”CCR2” has been modified to ”CCL2 receptor”.
  3. Line 176-178: ”After MSCs treatment in stroke mice, there was significant increase and inhibition in the expression of the axonal growth-associated protein and the axonal growth-inhibiting proteins, respectively” has been modified to ”After MSCs treatment in stroke mice, there were a significant increase in the expression of the axonal growth-associated proteins and a significant decrease in the expression of the axonal growth-inhibiting proteins”.
  4. Line 183: ”Third” has been modified to ”Besides”.
  5. Line 196-197 “the density of bicocorticoids, axons, myelin sheaths, and nerve filaments” has been modified to ”doublecortin, axon, myelin and neurofilament density'.
  6. Line 202: ”By” has been modified to ”by”.
  7. Line 203: ”First” has been removed.
  8. Line 213: ”research” has been modified to ”researches”.
  9. Line 262-263: ”the function and differentiation” have been modified to ”the function”.
  10. Line 366: ”N.-J.D., H.-L.H., S.-B.Z.” has been modified to ”N.-J.D., H.-L.H., S.-B.Z., J.W.”.
  11. Line 369: “Basic Research Program of Shanghai (20JC1412200)" has been put in the first place.

We hope that these changes have made the manuscript acceptable for publication. We owe a great deal to the editor for their patience and assistance in helping us to improve the paper.

Best regards,

Jiao Wang
